# ENHANCING DIVERSITY AND NOVELTY IN TEXT GENERATION VIA MULTI-VIEW EMBEDDINGS

**Arash Lagzian[1], Srinivas Anumasa[1], Dianbo Liu[1]**
[1]National University of Singapore
`alagzian@visitor.nus.edu.sg`
`{srinu_pd, dianbo}@nus.edu.sg`

## ABSTRACT

Large Language Models (LLMs) demonstrate remarkable proficiency in generating accurate and fluent text. However, they often struggle with diversity and novelty, leading to repetitive or overly deterministic responses. These limitations stem from constraints in training data, including gaps in specific knowledge domains, outdated information, and an over-reliance on textual sources. Such shortcomings reduce their effectiveness in tasks requiring creativity, multi-perspective reasoning, and exploratory thinking. To address this challenge, we introduce multi-view embeddings, a novel approach that enriches input prompts with diverse perspectives derived from both textual and visual sources. By incorporating additional contextual information, this method enhances the variety and creativity of generated outputs. Importantly, our approach is model-agnostic, requiring no architectural modifications and being compatible with both open-source and proprietary LLMs. Furthermore, we propose a comprehensive evaluation framework that simultaneously measures diversity, novelty, and correctness—a first-of-its-kind methodology for assessing these three crucial aspects of LLM-generated content. We evaluate our method and framework on over 469,000 generated outputs from various well-known LLMs, demonstrating significant improvements in output diversity and novelty while maintaining quality and relevance. Our approach provides a scalable and practical solution for enhancing LLM performance across a wide range of applications, including brainstorming, creative writing, and multiple-choice question generation.

## 1 INTRODUCTION

Rapid advances in large language models (LLMs) have spurred an ongoing debate on the usefulness of these models on tasks that require human-level creativity. On the one hand, there are works that highlight the strengths of LLMs in creative writing Pépin et al. (2024); Orwig et al. (2024), poetry generation Porter & Machery (2024), idea generation Lee & Chung (2024); Si et al. (2024) and even originality Guzik et al. (2023). On the other hand, some studies suggest that LLM creativity is significantly weaker than human creativity Chakrabarty et al. (2024) and that LLM-generated stories exhibit observable shortcomings Sato (2023); Levenson (2023). A recent user study found that, the use of an AI assistant in writing appears to enhance the creativity of individual writers, however, it reduces the collective diversity of novel content produced by multiple writers Doshi & Hauser (2024). This suggests that we should examine the distribution of LLM creations for a given prompt instead of each creation individually. Language models like GPT-4o OpenAI (2023) can produce repetitive answers in story telling and the answer not the same as original text Xu et al. (2024).

Existing methods primarily rely on single view prompts to generate responses, often resulting in limited diversity. To address this, we propose leveraging multiple views for prompting to enhance response diversity. However, ensuring the correctness of these diverse responses is equally crucial. Recognizing the importance of both novelty and accuracy, we introduce a novel framework designed to generate diverse and novel responses while also providing a mechanism to evaluate these aspects. Our key contributions in this work are as follows:

- We propose an architecture-independent approach to enrich generated text in terms of both novelty and diversity. By incorporating multiple views of the text embedding or image embedding in the case of image to text, our method encourages the model to produce more diverse and novel outputs.

- We introduce a framework to quantitatively assess generated responses based on diversity, novelty, and correctness.

- We conduct extensive experiments to demonstrate the effectiveness of our approach, evaluating over $469K$ generated responses and showcasing its improvements over existing LLM models.

## 2 RELATED WORK

Creative writing Kobak et al. (2024); Lee & Chung (2024) is on the rise; however, some studies suggest that content generated by human users tends to be more creative Kefford (2023). This study show that ChatGPT's ideas are more purchased from Wharton MBA students. There is an ongoing debate about whether LLMs can enhance creativity. To explore this, Lee & Chung (2024) demonstrates that when participants were tasked with generating creative ideas for everyday purposes, their creativity improved. However, Begus (2023) finds that AI-generated narratives often lack imagination and typically include plot twists in a more casual manner. Additionally, Chakrabarty et al. (2024) invited expert writers to evaluate stories generated by LLMs versus those created by professional writers using a standard creativity test. Their findings suggest that LLM-generated stories are less creative compared to those written by professionals. Empirical studies have underscored this issue. For example, Si et al. (2024) conducted qualitative analyses involving human judgment and found that after generating 500 samples, 50% were non-repetitive ideas. However, in the following 1,500 generations, only an additional 50% of non-repetitive ideas were produced. Alarmingly, in the final 2,000 rounds, just 12.5% of the generated ideas were non-repetitive. This suggests that while an individual LLM output may appear novel, when generating multiple outputs, the LLM tends to become repetitive, lacking the diversity necessary to effectively enhance collective creativity. This decline underscores the resource inefficiency and diminishing returns in prolonged LLM-generated content.

McCoy et al. (2023) suggests that novelty in LLM outputs can be detected by ensuring "the text must not have been copied from the training data." However, a more recent study by Xu et al. (2024) argues that this definition is superficial. In their experiment on story continuation, they demonstrate that while GPT-4-generated samples may meet this standard, the generated continuations are still quite conventional and lack diversity. Shaib et al. (2024) analyze different existing scores that can help measure diversity in LLM outputs, but these metrics all focus on surface-level features such as n-gram overlaps. Ghosal et al. (2022) indicate that "identifying novel text is not straightforward because the text many have less lexical overlap yet convey the same information." and to the best of our knowledge there is no study to evaluate the diversity, novelty, and correctness of the generated outputs at the same time.

## 3 PROPOSED METHOD

### 3.1 MULTI-VIEW EMBEDDING

The inability of existing LLM models to generate diverse and novel text persists even after fine-tuning the temperature parameter. We propose that instead of solely adjusting the temperature, prompting models from multiple perspectives can effectively encourage the generation of more diverse and novel text. Figure 1a offers an overview of our approach. Instead of directly interacting with the LLM model to generate a response, we first interact with a multi-view generator to create several perspectives of the given prompt. These generated views are then fed into the LLM model to produce the final response. The following section will give a more detailed overview of our approach. In sections 3.1.1 and 3.1.2, we explain the concepts of text view embedding and image view embedding within our method. Our experiments show that this approach enhances the model's understanding, increases output diversity, and boosts creativity.

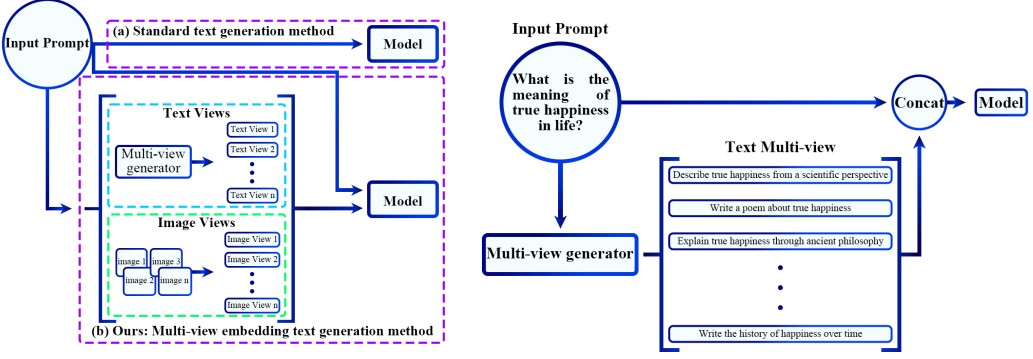

(a) Comparison between existing text generation methods and our method.

(b) Text Multi-View Embedding.

Figure 1: (a) Comparison of existing text generation methods and our method. (b) Illustration of the Text Multi-View Embedding.

### 3.1.1 TEXT VIEW EMBEDDING

Text Multi-View Embedding enhances the input prompt by generating multiple diverse perspectives or representations of the same concept, which are then combined before being fed into the model. This approach aims to provide a more comprehensive and context-rich input using various textual sources. These sources can be gathered from the internet, added manually, or even generated by a language model. In this work, we utilize GPT-4o as our text multi-view generator, with all texts in English. Figure 1b illustrates the text view embedding section of our method. In Lau et al. (2024), GPT-4o was also used to extract diverse perspectives from math questions, resulting in improved performance on reasoning tasks.

### 3.1.2 IMAGE VIEW EMBEDDING

In addition to Text Multi-View Embedding, we also introduce Image Multi-View Embedding. Images contain rich contextual information and, unlike text, can offer diverse and multiple perspectives. Image Multi-View Embedding is used to enhance the input prompt by incorporating multiple image-based perspectives. This method starts by crawling for images related to the input prompt, which serve as visual representations of the concept. Once relevant images are retrieved, the Qwen-2VL Wang et al. (2024) vision-language model is used to describe each image. These descriptions capture the visual content in textual form; however, they may lack consistency in writing style or contain structural issues. To improve the quality and coherence of these descriptions, we use a language model, in this case GPT-4O-Mini, to rewrite and refine the original descriptions. The refined descriptions ensure that the textual representation of visual content is well-structured and stylistically consistent. Instead of directly concatenating the descriptions with the input prompt, we use the refined descriptions as additional context when generating the final response. The model utilizes the detailed information from the rewritten image descriptions to provide richer, more accurate answers to the input prompt. The process of image view embedding is shown in the Figure 2.

This approach increases the semantic richness and contextual awareness of the input. Visual content often highlights aspects that textual prompts alone might overlook, making it possible to generate more comprehensive and creative outputs. The rewriting process ensures high-quality input, which enhances the overall coherence and informativeness of the model's responses. Figure 4 in appendix A shows an example of an image obtained by crawling over the internet using the input prompt, with its corresponding Image view and Answer.

### 3.2 METRICS

To evaluate the responses generated by different LLMs for a given input prompt, it is crucial to consider multiple aspects of the generated text in order to quantify the model's performance. Existing works typically focus on one aspect—such as novelty, diversity, or correctness—individually. In

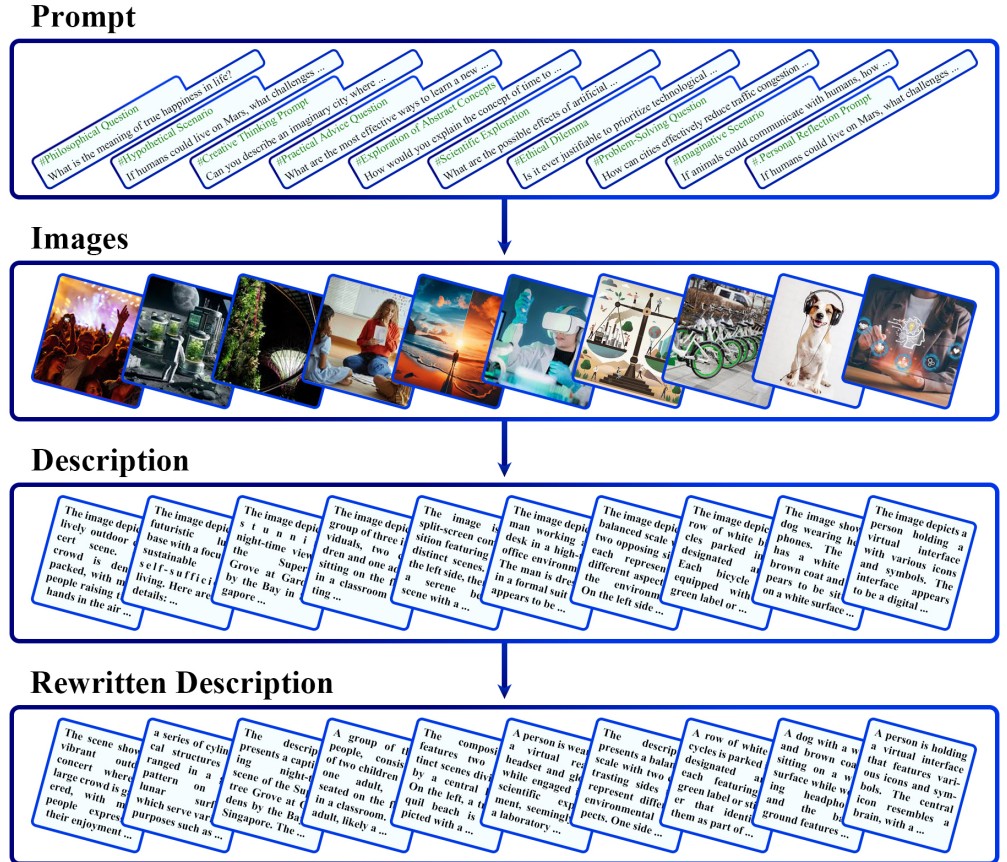

Figure 2: This Figure illustrates the process of preparing image view embeddings and provides an example for 10 input prompts. Row 1 displays 10 prompts from various subjects. Row 2 shows images crawled based on each input prompt. Row 3 presents the descriptions generated for the images, and Row 4 contains the rewritten descriptions, which serve as our image view embeddings.

contrast, our work takes a comprehensive approach by considering all these aspects to assess the model's performance.

### 3.2.1 DIVERSITY MEASURE

The diversity in the text generated by LLMs can be measured from various aspects. Two of the most important aspects are:

- Diversity across different text responses.
- Diversity within different tokens generated in a single text response.

In this work, first we try to describe each metric and then plot the importance of each one in diversity measurement and special cases for each metric and finally talk about the results.

**MTLD:** Lexical diversity is a key measure of linguistic richness in generated text. To evaluate this, we use the Measure of Textual Lexical Diversity (MTLD) McCarthy & Jarvis (2010). MTLD calculates the mean length of text segments that maintain a predefined type-token ratio (TTR). This approach overcomes limitations of traditional TTR metrics, which are sensitive to text length. Following McCarthy & Jarvis (2010), we set the TTR threshold to 0.72, as this value has been empirically validated to balance sensitivity and robustness across a variety of text datasets.

**Semantic Diversity of Text (SDT)**   Semantic diversity based on Term Frequency-Inverse Document Frequency (TF-IDF) captures the variation in term usage relative to the importance of terms across multiple outputs. By assigning weights to terms based on their frequency and distinctiveness, TF-IDF provides insights into the uniqueness of generated content. High semantic diversity is reflected in less repetition of terms and greater differentiation in word usage across outputs. This metric is instrumental in evaluating language models' ability to generate contextually diverse and meaningful text, which is particularly valuable in creative and informative content generation tasks.

Given a set of responses $r_j$ for a given $j^{th}$ prompt. The *TF-IDF* score for a term $t$ in response $r_j$ is calculated as:

$$\text{TF-IDF}(t, r_j) = \text{TF}(t, r_j) \times \text{IDF}(t)$$

where The Term Frequency for a term $t$ in the set of responses $r_j$ is calculated as: $\text{TF}(t, r_j) = \frac{\text{Number of times term } t \text{ appears in } r_j}{\text{Total number of terms in } r_j}$ and the Inverse Document Frequency for a term $t$ is calculated as: $\text{IDF}(t) = \log\left(\frac{m}{\text{DF}(t)}\right)$. Where, $m$ is the total number of prompts and $\text{DF}(t)$ is the number of set of responses that contain the term $t$. To measure the *semantic diversity*, we aggregate the *TF-IDF* scores of all terms in all the set of responses.

$$\textbf{SDT} = \frac{1}{m}\sum_{j=1}^{m}\frac{1}{n_j}\sum_{i=1}^{n_j}\text{TF-IDF}(t_i^j, r_j)$$

**Semantic Diversity of Embeddings (SDE)**   Semantic diversity using BERT measures the variation in semantic content across generated text outputs. Unlike lexical diversity, which focuses on surface-level differences in word usage, BERT-based semantic diversity captures deeper contextual differences by embedding sentences into a dense semantic space. We compute pairwise cosine similarity between answers embeddings and subtract mean value from one, so higher semantic diversity, reflecting a greater range of meaning and contextual richness in our setting. This approach leverages the contextualized representations of BERT Devlin et al. (2019), offering a robust and nuanced evaluation of the diversity and coherence of model-generated content.

The **SDE** of embeddings can be quantified by calculating the average pairwise *cosine distance* between the embeddings of the responses. Given a set of embeddings $\{\mathbf{e}_1, \mathbf{e}_2, \ldots, \mathbf{e}_n\}$, the cosine similarity between two embeddings $\mathbf{e}_i$ and $\mathbf{e}_j$ is defined as: $\text{Cosine Similarity}(\mathbf{e}_i, \mathbf{e}_j) = \frac{\mathbf{e}_i \cdot \mathbf{e}_j}{\|\mathbf{e}_i\|\|\mathbf{e}_j\|}$ The cosine distance is then can be computed as: $1 - \text{Cosine Similarity}(\mathbf{e}_i, \mathbf{e}_j)$ The **SDE** of a set of embeddings is then defined as the average cosine distance between all pairs of embeddings:

$$\textbf{SDE} = \frac{2}{n(n-1)}\sum_{1 \leq i < j \leq n}\left(1 - \frac{\mathbf{e}_i \cdot \mathbf{e}_j}{\|\mathbf{e}_i\|\|\mathbf{e}_j\|}\right)$$

**Self-BLUE**   Self-BLEU evaluates the diversity of generated text by measuring how similar each generated sample is to the others. Unlike traditional BLEU, which compares a generated sample to reference texts, Self-BLEU treats each sample as a reference for the others. A high Self-BLEU score signals redundancy, reflecting repeated patterns or limited variability Zhu et al. (2018). In our setting we subtract the mean value across all answers from 1 to measure the diversity across all answers.

**Lexical Entropy**   It quantifies the uncertainty or variability in word choice within generated text, serving as a measure of lexical diversity. Higher entropy indicates a broader and more varied vocabulary, while lower entropy suggests repetitive or predicTable word usage. We calculate Shannon entropy (in bits) over all tokens in the generated responses, based on their relative frequencies. For a given set of responses, the probability of each unique token is determined, and the entropy is computed as: $H = -\sum_i p_i \log_2 p_i$ where $p_i$ represents the probability of the $i$-th token. This metric captures how evenly distributed word usage is across the text. A higher lexical entropy signifies reduced repetition and a richer vocabulary, making it a valuable indicator of linguistic creativity in language models.

### 3.2.2   NOVELTY MEASURE

Novelty detection plays a crucial role in evaluating and improving the output of large language models (LLMs), especially in distinguishing existing knowledge (premise set) from new, previously

unknown information (hypothesis). This task can be formulated as a Natural Language Inference (NLI) problem, where the goal is to determine whether a candidate hypothesis introduces novel information not contained in the premise set. Prior research has explored multiple approaches for novelty detection, including leveraging NLI frameworks and measuring semantic similarity using embedding-based models like SBERT Ghosal et al. (2022).

In our work, we wanted to assess the inherent capability of LLMs to perform as NLI models in detecting novel versus non-novel documents. To achieve this, we used **TAP-DNLD 1.0** Ghosal et al. (2018) a dataset consisting of around $2.8k$ novel and $2.7k$ non-novel documents, and each document belongs to one of the ten categories and each target document labeld versus three source document by human. For comparison , first we sampled from this dataset in three different seeds and in each seed from each category sampled 5 novel and 5 non-novel document to have a balance data and totally 300 document then We used GPT-4o and SBERT as novelty detector to have better comparison and results reported in Table 1.

Table 1: Classification metrics to evaluate novelty detectors GPT-4o and SBERT models on dataset **TAP-DNLD 1.0** Ghosal et al. (2018).

| Seed | Model | Accuracy | Precision | Recall | F1-score |
|---|---|---|---|---|---|
| Seed0 | SBERT | 0.6700 | 0.6269 | 0.8400 | 0.7179 |
| | GPT-4O | **0.6900** | **0.6415** | **0.8600** | **0.7347** |
| Seed1 | SBERT | **0.7200** | **0.6528** | **0.9400** | **0.7705** |
| | GPT-4O | 0.6600 | 0.6086 | 0.9000 | 0.7260 |
| Seed2 | SBERT | 0.6900 | 0.6338 | 0.9000 | 0.7438 |
| | GPT-4O | **0.6950** | **0.6368** | **0.9100** | **0.7492** |

After establishing the baseline performance of these models, we utilized GPT-4o and SBERT as novelty detectors to evaluate the novelty of generated outputs across different language models, as well as to assess the impact of our method on the novelty of generated responses. The novelty detection process is illustrated in Figure 3. In this approach, for each set of prompt outputs, the first generated response is considered novel by default. Subsequent responses are sequentially analyzed by the novelty detector, where each new answer is compared against previously identified novel responses. This comparison is conducted in a hypothesis-premise framework, where the new response serves as the hypothesis, and previously identified novel answers form the premise set. If the new answer introduces additional information not present in the premise set, it is classified as a novel answer and added to the set. Otherwise, it is labeled as redundant.

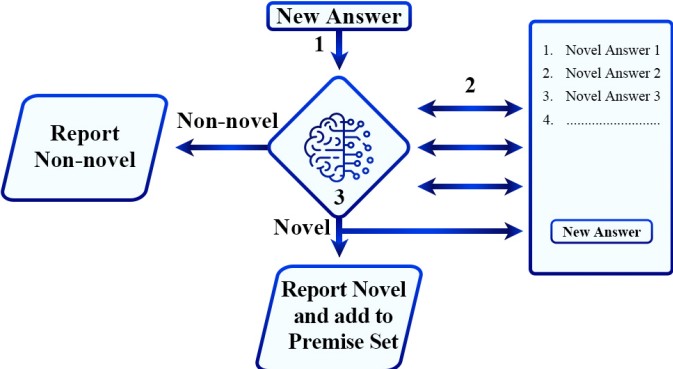

Figure 3: Novelty detection method by using GPT-4o and SBERT.

### 3.2.3 CORRECTNESS MEASURE

Answers could be novel or even diverse but completely irrelevant to the input prompt, or they can have different structures. In this section we want to answer these two questions:

- Is the generated answer correct and relevant to the given input prompt?

  • How well does the generated answer adhere to proper language structure and grammatical accuracy?

To address the first question, we designed an experiment to assess whether language models can accurately detect relevant answers to a given prompt. For this evaluation, we used the GPT-WritingPrompts dataset Huang et al. (2024), which contains approximately 97K unique prompts along with responses from both humans and GPT-3.5. However, in our study, we exclusively utilized the human-generated answers. We sampled data using ten different random seeds, selecting 1K samples per seed. These samples were then clustered into ten distinct groups. From each cluster, we randomly picked 100 prompts. For each selected prompt, we assigned one relevant answer from the same cluster, labeling it as "Correct," and one irrelevant answer from a different cluster, labeling it as "Incorrect," ensuring that the chosen answers were distinct. Since the answers are approximately 500 words long, we assume that if an answer is related to the prompt, its summary will also be relevant, and similarly, if an answer is irrelevant, its summary will remain unrelated. Based on this assumption, we use the GPT-4o model to generate summaries for both correct and incorrect answers, setting the maximum length to 250 words. Next, we evaluate the ability of two well-known LLMs, GPT-4o and DeepSeekV3 DeepSeek-AI et al. (2024), to determine the correctness of the answers. The results, presented in Table 2, indicate that GPT-4o outperforms DeepSeekV3 in detecting correctness. An example prompt and its related responses are provided in Appendix B. Based on these findings, we use GPT-4o for correctness detection in our experiments.

Table 2: Results on classification task to detect correct answer across all seeds for DeepSeekV3 and GPT-4O models. Complete results can be found in Table 12, Appendix A.

| Model | Accuracy | Precision | Recall | F1-score |
|---|---|---|---|---|
| DeepSeekV3 | 0.8410 ± 0.0223 | **0.9564 ± 0.0285** | 0.7270 ± 0.0447 | 0.8191 ± 0.0301 |
| GPT-4O | **0.8980 ± 0.0214** | 0.9468 ± 0.0255 | **0.8310 ± 0.0316** | **0.8946 ± 0.0228** |

To address the second question, we designed an experiment to evaluate language models based on their ability to assign scores to generated English texts. For this evaluation, we utilized the IELTS Writing Task 2 dataset with labeled scores Mazlumi (2023) from Kaggle. This dataset contains 642 responses for Task 1 and 793 responses for Task 2. Since only Task 2 is relevant to our study, we used its questions and answers to assess language models. We randomly sampled 100 responses in three different seeds, conducting two iterations to evaluate text quality. Two well-known models assigned scores between 1 and 9, and the results are presented in Table 3. Based on this experiment, we selected the DeepSeekV3 model to evaluate responses in terms of grammar and overall English structure, assigning each answer a score between 1 and 10. The prompt templates for both experiments are provided in Appendix B. Based on this experiment we used DeepSeekV3 model to evaluate the generated outputs.

Table 3: MSE (Mean ± Std) for GPT-4O and DeepSeekV3 per seed for two iterations. Lower values are bolded.

| Seed | GPT-4O | DeepSeekV3 |
|---|---|---|
| Seed0 | 2.3775 ± 0.0350 | **2.0775 ± 0.0025** |
| Seed1 | 2.2288 ± 0.0512 | **2.0300 ± 0.0050** |
| Seed2 | 2.1562 ± 0.0587 | **1.9050 ± 0.0500** |

## 4 EXPERIMENTAL RESULTS

After introducing various diversity metrics in Section 3.2.1, we designed an experiment to assess the diversity of different language models. Additionally, we applied the multi-view embedding method to three open-source models—GPT-2 Medium Radford et al. (2019), Qwen2.5-1.5B Team (2024), and DeepSeek-R1-7B DeepSeek-AI et al. (2025)—as well as two API-based models, GPT-4o OpenAI (2023) and GPT-4o Mini OpenAI (2024). Our experiments utilized 10 diverse prompts

spanning multiple domains (provided in Appendix B), instructing each model to generate responses. This process was repeated across different sample sizes ranging from 100 to 1500, with a fixed maximum sequence length of $125$, resulting in a total of $469k$ generated responses and for having fair compression we used the same parameters for all models like $tempreture = 0.9$ and $top\_k = 0.95$. The generated outputs were then evaluated in terms of diversity, novelty, and correctness. Table 4 presents the diversity measurement results. The experimental findings demonstrate that our method enhances diversity across all models. In some cases, we observed up to a threefold increase in the diversity of generated outputs.

Table 4: Mean ± standard deviation of diversity metrics across five different sample sizes ranging from 100 to 1500 per each prompt (10 prompts), with $max\_length = 125$. Detailed results are provided in Appendix A.

| Models | MTLD | TF-IDF | Lexical Entropy | Semantic Embedding | Self-BLEU |
|---|---|---|---|---|---|
| GPT-2 | **41.36 ± 0.05** | 0.86 ± 0.01 | 8.14 ± 0.09 | 0.55 ± 0.00 | 0.61 ± 0.10 |
| GPT-2 + Text View | 35.27 ± 0.99 | **0.89 ± 0.01** | **8.33 ± 0.13** | **0.64 ± 0.01** | **0.67 ± 0.08** |
| Qwen | 67.17 ± 0.15 | 0.80 ± 0.01 | 7.90 ± 0.05 | 0.22 ± 0.00 | 0.27 ± 0.08 |
| Qwen + Text View | 67.63 ± 0.11 | **0.86 ± 0.01** | **8.45 ± 0.07** | **0.39 ± 0.00** | **0.41 ± 0.09** |
| Qwen + Image View | **70.29 ± 0.04** | 0.84 ± 0.01 | 8.28 ± 0.07 | 0.30 ± 0.00 | 0.38 ± 0.09 |
| GPT-4O Mini | 57.95 ± 0.01 | 0.63 ± 0.01 | 7.06 ± 0.02 | 0.11 ± 0.00 | 0.10 ± 0.05 |
| GPT-4O Mini + Text View | **59.61 ± 0.12** | **0.80 ± 0.04** | **7.92 ± 0.04** | **0.33 ± 0.01** | **0.26 ± 0.09** |
| GPT-4O Mini + Image View | 58.71 ± 2.03 | 0.79 ± 0.02 | 7.80 ± 0.04 | 0.30 ± 0.01 | 0.25 ± 0.08 |
| GPT-4O | 57.46 ± 0.47 | 0.66 ± 0.01 | 7.00 ± 0.16 | 0.11 ± 0.02 | 0.09 ± 0.04 |
| GPT-4O + Text View | **58.25 ± 0.04** | 0.78 ± 0.01 | **7.85 ± 0.06** | **0.31 ± 0.01** | 0.25 ± 0.08 |
| GPT-4O + Image View | 55.09 ± 0.31 | **0.79 ± 0.01** | 7.77 ± 0.04 | **0.31 ± 0.00** | **0.29 ± 0.09** |
| DeepSeek-R1 | 52.49 ± 0.11 | 0.79 ± 0.01 | 7.63 ± 0.06 | 0.24 ± 0.00 | 0.26 ± 0.08 |
| DeepSeek-R1 + Text View | **54.62 ± 0.20** | **0.85 ± 0.01** | **8.19 ± 0.10** | **0.39 ± 0.01** | **0.39 ± 0.09** |
| DeepSeek-R1 + Image View | 54.35 ± 0.05 | 0.83 ± 0.01 | 8.06 ± 0.08 | **0.39 ± 0.00** | 0.36 ± 0.09 |

Table 5 illustrates the novelty score of two well-known large language models GPT-4o and DeepSeek-R1 and impact of our method. By incorporating multi-view embeddings, we enriched the input representation, leading to the generation of more novel and diverse responses from these models. This analysis provides valuable insights into how multi-view embedding strategies influence novelty detection and enhance the creativity of LLM outputs. The results show that for GPT-4o, our approach in some cases led to around ninefold improvement in novelty when text or image view embeddings were applied. Similarly, for DeepSeek-R1—one of the most inherently creative models—our method resulted in approximately a twofold increase in the novelty score. Additional results for other models, evaluated solely using SBERT as the novelty detector, are presented in Table 13 in Appendix A.

The evaluation of answer correctness across different models is presented in Table 6, considering two aspects introduced in Section 3.2.3. In Table 6a, we assess the correctness of generated answers based on their relevance to the input prompt, using GPT-4o as the correctness evaluator. The results indicate that all models achieve high correctness scores in this aspect. Additionally, Table 6b evaluates the correctness of three language models from an English language structure perspec-

Table 5: Results on percentage of novelty score across different models according to two novelty detectors GPT-4o and SBERT. More results for another models exist in Table 13.

| | num_samples = 100 | | num_samples = 250 | num_samples = 500 |
|---|---|---|---|---|
| Model | GPT-4o | SBERT | SBERT | SBERT |
| GPT-4o | 10.60 | 5.4 | 3.52 | 2.9 |
| GPT-4o + Text View | 29.30 | 40.2 | **32.08** | **24.48** |
| GPT-4o + Image View | **42.60** | **47.3** | 30.44 | 22.54 |
| DeepSeek-R1 | 27.40 | 43.6 | 36.24 | 31.82 |
| DeepSeek-R1 + Text View | **46.20** | **75.3** | 63.88 | 55.86 |
| DeepSeek-R1 + Image View | 40.60 | 73.7 | **65.04** | **56.18** |

tive. The results show that GPT-2 exhibits lower correctness scores in its generated answers, while the other two well-known LLMs perform significantly better. This evaluation was conducted using DeepSeekV3 as the correctness evaluator.

(a) correctness results (%)

| Model | GPT-4o |
|---|---|
| GPT-4o mini | 99.81 |
| GPT-4o mini + Text View | 91.00 |
| GPT-4o mini + Image View | 87.00 |
| Qwen | 93.77 |
| Qwen + Text View | 76.60 |
| Qwen + Image View | 82.50 |
| GPT-4o | 99.60 |
| GPT-4o + Text View | 92.60 |
| GPT-4o + Image View | 94.60 |
| DeepSeek-R1 | 91.80 |
| DeepSeek-R1 + Text View | 81.00 |
| DeepSeek-R1 + Image View | 53.90 |

(b) Mean score (1 to 10) for correctness from English structure aspects.

| Model | DeepSeekV3 |
|---|---|
| GPT-2 | 3.29 |
| GPT-2 + Text View | 2.40 |
| GPT-4o | 8.07 |
| GPT-4o + Text View | 8.05 |
| GPT-4o + Image View | 8.06 |
| DeepSeek-R1 | 7.96 |
| DeepSeek-R1 + Text View | 7.15 |
| DeepSeek-R1 + Image View | 6.10 |

Table 6: (a) and (b) show correctness results for different evaluation aspects for $num\_sample = 100$ per prompt (10 prompts).

## 5 ABLATION STUDY

While our method significantly enhances the diversity and novelty of generated outputs across different language models, we conducted an ablation study to examine its impact on correctness and English language structure. The results indicate that increasing output diversity and novelty can sometimes lead to a decline in correctness and grammatical quality. In particular, as shown in Table 6, applying multi-view embeddings led to a slight decrease in correctness scores, especially for models that originally produced highly relevant responses. This suggests that while the model explores a broader range of responses, it may deviate from the most precise or expected answer. Similarly, Table 6b demonstrates that the grammatical correctness of responses is also affected, with models generating more varied but less structurally refined answers. These findings highlight the trade-off between novelty and correctness, a well-known challenge in text generation. While increasing diversity enriches the model's expressiveness and reduces repetitiveness, it can also introduce syntactic errors or responses that, while novel, may not align perfectly with human expectations.

## 6 CONCLUSION

In this work, we present multi-view embedding, a model-agnostic approach that enriches the input prompt with diverse textual and visual sources to enhance the diversity and novelty of generated responses for both open-source and closed-source LLMs. Additionally, we introduce a framework to assess the diversity, novelty, and correctness of the generated outputs from LLMs. We validate both our method and framework using real-world datasets and $469k$ generated answers from various LLMs.

## 7 FUTURE WORK

As demonstrated in this paper, incorporating textual or visual views enhances the output diversity and creativity of existing language models. In our future work, we aim to explore the impact of additional view types, such as videos, songs, maps, or even mathematical equations, as multi-view embeddings on model performance. Furthermore, we plan to extend our proposed framework to assess not only diversity, novelty, and correctness, but also effectiveness and usefulness, in order to provide a more comprehensive evaluation of the behavior of large language models.

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

# A APPENDIX: DETAILED EXPERIMENTAL RESULTS

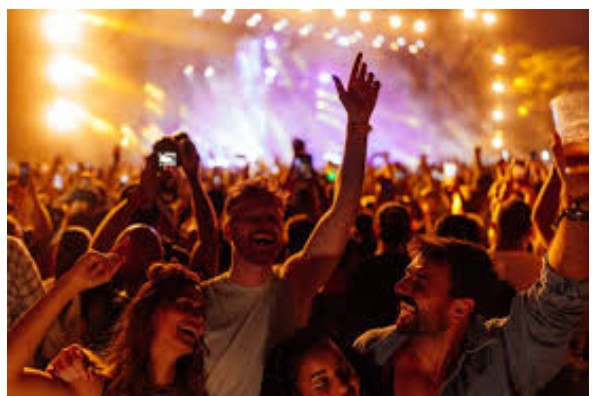

**Prompt:** *What is the meaning of true happiness in life?*

**Image View:** *The scene shows a vibrant outdoor concert where a large crowd is gathered, with many people expressing their enjoyment by raising their hands in the air. Bright lights illuminate the area, particularly focusing on the stage in the background. The audience is lively and energetic, with some individuals recording the experience on their phones. The atmosphere is filled with excitement and celebration, characteristic of a live music event.*

**Answer:** *True happiness in life often encompasses experiences that bring joy, connection, and personal growth through various activities such as participating in music concerts like the one described. Engaging in shared experiences with others creates bonds and enriches our social lives. Music, being an art form rooted deeply in human culture, has long been associated with positive emotions; it can evoke strong feelings of joy, community, and self-expression. In such a vibrant outdoor concert setting, the collective emotion displayed by the attendees—raising their hands in the air, indicating enthusiasm, and enthusiastically recording the moment with smartphones—are vivid expressions of true happiness and engagement.*

Figure 4: An example of Image view related to the input prompt and answered by Qwen2.5-1.5B model.

Table 7: Diversity Metrics for different models. Higher values indicate better performance. $num\_samples = 100$, $max\_length = 125$

| Models | MTLD | TF-IDF | Lexical Entropy | Semantic Embedding | Self-BLEU |
|---|---|---|---|---|---|
| GPT-2 | **41.44** | 0.85 | 7.98 | 0.54 | 0.76 |
| GPT-2 + Text View | 35.91 | **0.88** | **8.11** | **0.64** | **0.80** |
| Qwen | 67.00 | 0.78 | 7.81 | 0.22 | 0.41 |
| Qwen + Text View | 67.77 | **0.85** | **8.32** | **0.39** | **0.56** |
| Qwen + Image View | **70.24** | 0.82 | 8.15 | 0.29 | 0.50 |
| GPT-4O Mini | 57.98 | 0.61 | 7.03 | 0.11 | 0.16 |
| GPT-4O Mini + Text View | 59.68 | **0.77** | **7.87** | **0.34** | **0.39** |
| GPT-4O Mini + Image View | **60.19** | **0.77** | 7.76 | 0.32 | 0.38 |
| GPT-4O | 57.17 | 0.64 | 6.85 | 0.10 | 0.15 |
| GPT-4O + Text View | **58.25** | 0.76 | **7.75** | **0.31** | 0.38 |
| GPT-4O + Image View | 54.66 | **0.77** | 7.68 | **0.31** | **0.44** |
| DeepSeek-R1 | 52.34 | 0.76 | 7.54 | 0.24 | 0.39 |
| DeepSeek-R1 + Text View | **55.07** | **0.84** | **8.07** | **0.40** | **0.52** |
| DeepSeek-R1 + Image View | 54.37 | 0.82 | 7.95 | 0.39 | **0.52** |

Table 8: Diversity Metrics for different models. Higher values indicate better performance. $num\_samples = 250$, $max\_length = 125$.

| Models | MTLD | TF-IDF | Lexical Entropy | Semantic Embedding | Self-BLEU |
|---|---|---|---|---|---|
| GPT-2 | **41.35** | 0.86 | 8.11 | 0.55 | 0.68 |
| GPT-2 + Text View | 35.69 | **0.89** | **8.28** | **0.64** | **0.72** |
| Qwen | 67.45 | 0.79 | 7.88 | 0.22 | 0.31 |
| Qwen + Text View | 67.68 | **0.85** | **8.43** | **0.39** | 0.47 |
| Qwen + Image View | **70.27** | 0.83 | 8.26 | 0.30 | **0.48** |
| GPT-4O Mini | 57.94 | 0.62 | 7.05 | 0.11 | 0.16 |
| GPT-4O Mini + Text View | 59.79 | **0.79** | **7.91** | **0.33** | **0.38** |
| GPT-4O Mini + Image View | **60.38** | 0.78 | 7.82 | 0.31 | 0.28 |
| GPT-4O | 57.40 | 0.66 | 6.88 | 0.09 | 0.10 |
| GPT-4O + Text View | **58.21** | 0.78 | **7.85** | **0.31** | 0.29 |
| GPT-4O + Image View | 55.23 | **0.79** | 7.77 | **0.31** | **0.34** |
| DeepSeek-R1 | 52.41 | 0.78 | 7.60 | 0.24 | 0.30 |
| DeepSeek-R1 + Text View | **54.44** | **0.85** | **8.12** | **0.40** | **0.45** |
| DeepSeek-R1 + Image View | 54.42 | 0.83 | 8.05 | 0.39 | 0.41 |

Table 9: Diversity Metrics for different models. Higher values indicate better performance. $num\_samples = 500$, $max\_length = 125$.

| Models | MTLD | TF-IDF | Lexical Entropy | Semantic Embedding | Self-BLEU |
|---|---|---|---|---|---|
| GPT-2 | **41.30** | 0.87 | 8.17 | 0.55 | 0.60 |
| GPT-2 + Text View | 35.77 | **0.90** | **8.36** | **0.64** | **0.65** |
| Qwen | 67.13 | 0.80 | 7.91 | 0.22 | 0.25 |
| Qwen + Text View | 67.62 | **0.86** | **8.48** | **0.39** | **0.40** |
| Qwen + Image View | **70.36** | 0.84 | 8.32 | 0.30 | 0.36 |
| GPT-4O Mini | 57.95 | 0.63 | 7.07 | 0.11 | 0.07 |
| GPT-4O Mini + Text View | 59.53 | **0.79** | **7.93** | **0.33** | **0.22** |
| GPT-4O Mini + Image View | **60.36** | **0.79** | 7.85 | 0.31 | 0.21 |
| GPT-4O | 56.99 | 0.67 | 6.89 | 0.09 | 0.07 |
| GPT-4O + Text View | **58.22** | **0.79** | **7.86** | 0.30 | 0.23 |
| GPT-4O + Image View | 55.29 | 0.79 | **7.79** | **0.31** | **0.27** |
| DeepSeek-R1 | 52.56 | 0.79 | 7.66 | 0.24 | 0.24 |
| DeepSeek-R1 + Text View | **54.52** | **0.86** | **8.22** | **0.39** | **0.37** |
| DeepSeek-R1 + Image View | 54.27 | 0.84 | 8.08 | **0.39** | 0.34 |

Table 10: Diversity Metrics for different models. Higher values indicate better performance. $num\_samples = 1000$, $max\_length = 125$.

| Models | MTLD | TF-IDF | Lexical Entropy | Semantic Embedding | Self-BLEU |
|---|---|---|---|---|---|
| GPT-2 | **41.33** | 0.87 | 8.21 | 0.54 | 0.53 |
| GPT-2 + Text View | 35.66 | **0.90** | **8.43** | **0.64** | **0.59** |
| Qwen | 67.10 | 0.81 | 7.94 | 0.22 | 0.20 |
| Qwen + Text View | 67.44 | **0.87** | **8.50** | **0.38** | **0.34** |
| Qwen + Image View | **70.29** | 0.85 | 8.34 | 0.30 | 0.30 |
| GPT-4O Mini | 57.95 | 0.64 | 7.08 | 0.11 | 0.05 |
| GPT-4O Mini + Text View | **59.57** | **0.88** | **7.95** | **0.33** | 0.17 |
| GPT-4O Mini + Image View | 55.76 | 0.81 | 7.79 | 0.29 | **0.19** |
| GPT-4O | 57.83 | 0.66 | 7.14 | 0.13 | 0.06 |
| GPT-4O + Text View | **58.29** | 0.79 | **7.90** | **0.31** | 0.18 |
| GPT-4O + Image View | 55.12 | **0.80** | 7.80 | **0.31** | **0.21** |
| DeepSeek-R1 | 52.54 | 0.79 | 7.67 | 0.24 | 0.19 |
| DeepSeek-R1 + Text View | **54.55** | **0.86** | **8.27** | **0.39** | **0.31** |
| DeepSeek-R1 + Image View | 54.38 | 0.84 | 8.11 | **0.39** | 0.28 |

Table 11: Diversity Metrics for different models. Higher values indicate better performance. $num\_samples = 1500$, $max\_length = 125$.

| Models | MTLD | TF-IDF | Lexical Entropy | Semantic Embedding | Self-BLEU |
|---|---|---|---|---|---|
| GPT-2 | **41.39** | 0.87 | 8.24 | 0.55 | 0.49 |
| GPT-2 + Text View | 33.30 | **0.90** | **8.47** | **0.66** | **0.58** |
| Qwen | 67.19 | 0.81 | 7.95 | 0.22 | 0.18 |
| Qwen + Text View | 67.64 | **0.87** | **8.52** | **0.38** | **0.30** |
| Qwen + Image View | **70.30** | 0.85 | 8.35 | 0.30 | 0.27 |
| GPT-4O Mini | 57.91 | 0.64 | 7.08 | 0.11 | 0.04 |
| GPT-4O Mini + Text View | **59.50** | **0.80** | **7.96** | **0.33** | 0.15 |
| GPT-4O Mini + Image View | 54.86 | 0.81 | 7.77 | 0.29 | **0.16** |
| GPT-4O | 57.90 | 0.66 | 7.14 | 0.13 | 0.05 |
| GPT-4O + Text View | **58.26** | 0.79 | **7.90** | 0.31 | 0.15 |
| GPT-4O + Image View | 55.14 | 0.80 | 7.82 | 0.31 | 0.18 |
| DeepSeek-R1 | **52.59** | 0.79 | 7.68 | 0.24 | 0.17 |
| DeepSeek-R1 + Text View | **54.53** | **0.86** | **8.28** | **0.39** | **0.28** |
| DeepSeek-R1 + Image View | 54.32 | 0.84 | 8.12 | **0.39** | 0.24 |

Table 12: Classification metricss for GPT-4O and DeepSeekV3 across different seeds to evaluate the ability of these models to detecet the correct answer.

| Seed | Model | Accuracy | Precision | Recall | F1-score |
|---|---|---|---|---|---|
| Seed0 | DeepSeekV3 | 0.8450 | **0.9367** | 0.7400 | 0.8268 |
| | GPT-4O | **0.9100** | 0.9271 | **0.8900** | **0.9082** |
| Seed1 | DeepSeekV3 | 0.8100 | **0.9198** | 0.6800 | 0.7816 |
| | GPT-4O | **0.8700** | 0.9111 | **0.8200** | **0.8632** |
| Seed2 | DeepSeekV3 | 0.8450 | 0.9726 | 0.7100 | 0.8187 |
| | GPT-4O | **0.9150** | **0.9663** | **0.8600** | **0.9091** |
| Seed3 | DeepSeekV3 | 0.7900 | 0.9265 | 0.6300 | 0.7500 |
| | GPT-4O | **0.8650** | **0.9195** | **0.8000** | **0.8556** |
| Seed4 | DeepSeekV3 | 0.8750 | 0.9000 | 0.7600 | 0.8588 |
| | GPT-4O | **0.9300** | **0.9778** | **0.8800** | **0.9268** |
| Seed5 | DeepSeekV3 | 0.8300 | 0.9342 | 0.7200 | 0.8068 |
| | GPT-4O | **0.9200** | **0.9773** | **0.8600** | **0.9149** |
| Seed6 | DeepSeekV3 | 0.8450 | **0.9859** | 0.7000 | 0.8187 |
| | GPT-4O | **0.8650** | 0.9011 | **0.8000** | **0.8556** |
| Seed7 | DeepSeekV3 | 0.8550 | **0.9863** | 0.7200 | 0.8324 |
| | GPT-4O | **0.9150** | 0.9462 | **0.8800** | **0.9123** |
| Seed8 | DeepSeekV3 | 0.8400 | **0.9857** | 0.6900 | 0.8118 |
| | GPT-4O | **0.8850** | 0.9753 | **0.7900** | **0.8729** |
| Seed9 | DeepSeekV3 | 0.8450 | **0.9600** | 0.7200 | 0.8229 |
| | GPT-4O | **0.8750** | 0.9032 | **0.8400** | **0.8750** |

Table 13: Results on percentage of novelty score across different models according to the novelty detector SBERT

| | num_samples = 100 | num_samples = 250 | num_samples = 500 |
|---|---|---|---|
| Model | SBERT | SBERT | SBERT |
| GPT2 | 98.10 | 96.76 | 95.46 |
| GPT2 + Text View | 99.70 | 98.88 | 98.72 |
| GPT-4o-mini | 5.71 | 3.44 | 2.50 |
| GPT-4o-mini + Text View | 47.50 | 32.24 | 24.30 |
| GPT-4o-mini + Image View | 47.30 | 30.44 | 18.70 |
| Qwen2.5-1.5B | 32.70 | 25.44 | 22.58 |
| Qwen2.5-1.5B + Text View | 68.70 | 59.60 | 52.64 |
| Qwen2.5-1.5B + Image View | 57.20 | 48.80 | 45.14 |

## B   APPENDIX: PROMPTS AND MORE INFORMATION

**Prompts used in this work**
We Used 10 prompts from different subjects to improve generalization on our results.
1.Philosophical Question:
"What is the meaning of true happiness in life?",

2.Hypothetical Scenario:
"If humans could live on Mars, what challenges would they face and
how could they overcome them?",

3.Creative Thinking Prompt:
"Can you describe an imaginary city where technology and nature
exist in perfect harmony?",

4.Practical Advice Question:
"What are the most effective ways to learn a new language
quickly?",

5.Exploration of Abstract Concepts:
"How would you explain the concept of time to someone who has
never experienced it?",

6.Scientific Exploration:
"What are the possible effects of artificial intelligence on
scientific research in the next decade?",

7.Ethical Dilemma:
"Is it ever justifiable to prioritize technological advancement
over environmental protection?",

8.Problem-Solving Question:
"How can cities effectively reduce traffic congestion without
compromising accessibility?",

9.Imaginative Scenario:
"If animals could communicate with humans, how would that change
our world?",

10.Personal Reflection Prompt:
"What qualities make someone a great leader, and how can those
qualities be developed?"

**Evaluate the correctness of prompt answer experiment details**
This is an example of a random prompt from seed 0 with it's correct answer and selected incorrect
answer from another cluster.  in Figure 5 the visualization of distance between two answers
demonstrated.

Prompt:
"In a isolated research facility , a researcher walks past a
janitor , only to stop later and realize that we do n't have a
janitor ."
"correct_answer":

"Something was off .  But something was always off in these
hallways .  The corridors slightly too long , slightly too narrow
for two people to walk comfortably abreast .  As if someone
had googled exactly what it was that made people unconsciously
uncomforTable and crammed every last wretched detail into the
place .  Shifting in the starched collar , he sniffed and pulled
the pen from the clipboard .  <newline> Trying to trace the events
that lead to him being here , it was nigh on impossible .  A
odd comment by a professor there , reading a specific page of a
newspaper , there was no defining reason that spurred him here
.  <newline> That said , no one could deny that it was a bold
career move , and a bold move for him in general .  He was in
Antarctica for Christ 's sake .  <newline> It brought him no
small joy that on slow days he would pace the corridors of the
facility , for lack of something better to do .  He had worked out
the optimal route based on path efficiency , and had divided his
routine between Surveying tasks so that the laps loosely resembled
a set – three reps of the uniformly grey corridors , spaced by
intervals of 45 minutes .  <newline> So on the second rep of the
routine , passing through a adjunct corridor connecting both polar
fieldwork hub A and a room masquerading as a laboratory , that he
saw a Figure that the very end of the hallway .  <newline> It 's
not my job to speculate on a person 's intelligence obviously .
But what I can say is it was a pretty damn long corridor and our
protagonist had enough time to realise that something was off .
Instead , the approach happened , he adopted his socially trained
response to the presence of an unknown but nonthreatening human
, an easy smile , and nodded as he went past the scruffy Figure
.  <newline> He took in some details .  The janitors uniform , one
that looked from the lack of utility more parodic that anything ,
as if dressed Halloween or something .  A slight gait , no hat ,
two buttons done up interrupted by the third undone and then the
fourth one buttoned up by the neck .  <newline> He rounded the
corner .  <newline> There was a pause .  <newline> He reprocessed
the moment .  <newline> Turning , he popped his head back around
the corner , and looked back down the corridor , visibly straining
.  <newline> 'Mom ?  '  <newline>"
"incorrect_answer":
"*God is everywhere .  * <newline> <newline> When I was a little
boy and my mom brought her friends home I used to go to the church
.  They used to keep it open all the time .  Back then churches
were n't afraid of thieves .  <newline> <newline> I used to sit
in the pews at night and then catch the morning sermon .  *God
is in all of us .  * used to say the priest .  His words would
wash over me , comforting and familiar .  I was part of something
great , all humanity connected by God 's love .  That 's what
started me on killing people too .  I always chose people for whom
salvation was a myth .  And I think I helped them .  Even if they
did n't know it .  <newline> <newline> Now I 've been chosen .  I
played god with people 's lives , now I am God .  I did n't bother
asking why .  It 's obvious .  <newline> <newline> Here I am now
.  Silently watching .  I will not punish the deserving .  I will
turn a blind eye when the innocents suffer .  The greedy , the
cheaters , the murderers and the rapists shall live good lives .
Their victims will with death upon themselves every day of their
lives .  The world will be an uglier place ."
"correct_summary":
"In the unsettling corridors of an Antarctic facility, designed
to evoke discomfort with their narrow and elongated design,

our protagonist reflects on how he ended up there. The journey to this remote place wasn't marked by a clear decision, but rather a series of subtle influences like passing comments and newspaper articles. Despite the oddity of his situation, relocating to such an isolated location was viewed as a bold career move. To combat the monotony, he developed a routine of pacing the dreary hallways, structuring his day around these efficient laps interspersed with his work.\n\n One day, as he completed his usual circuit and passed through a connecting corridor, he noticed a Figure at the far end. Instinctively, he reacted with a default nod and smile, a social conditioning when encountering strangers. Something about the person seemed odd, their janitorial uniform appeared almost like a parody, missing elements of practicality, and the person walked with a slight limp. After rounding a corner, the protagonist paused, puzzled, and reviewed the encounter in his mind. Curiosity piqued, he turned back, peering down the corridor with noticeable effort to see the Figure once more. The moment replays, and to his shock, he exclaims, M̈om?s̈uggesting the surprising recognition of his mother in this unexpected place.",
"incorrect_summary":
"The narrator reflects on their childhood experiences at church, where they found comfort in a priest's message that G̈od is in all of us.T̈his message led them to view humanity as interconnected through God's love. However, this belief also fueled their justification for killing, targeting those they deemed without hope of salvation. The narrator believed they were aiding these individuals, whether or not they realized it. Now, they claim to have ascended to a god-like status, implying they have the power to determine others' fates without questioning why they have been chosen for this role. The narrator adopts a detached and indifferent stance towards justice and morality, deciding not to punish those who might deserve it and ignoring the suffering of innocent people. They suggest that they will allow wrongdoers to thrive while their victims live in misery, predicting that the world will become a more üglyp̈lace as a result."
"prompt_cluster":
6
"incorrect_cluster":
7

**Template to evaluate the IELTS score prediction ability**
You are an IELTS examiner. Please evaluate the following essay and give a score between 1.0 and 9.0 based on the IELTS Writing Band Descriptors. The essay should be scored based on Task Response, Coherence and Cohesion, Lexical Resource, and Grammatical Range and Accuracy.
Question: {question}
Essay: {essay}
Please provide only a score between 1.0 and 9.0.

**Template to get the summary of prompt answers**
Please summarize the following text in no more than {max_words} words:
{text}

**Template to evaluate the correctness of answers**
PROMPT:

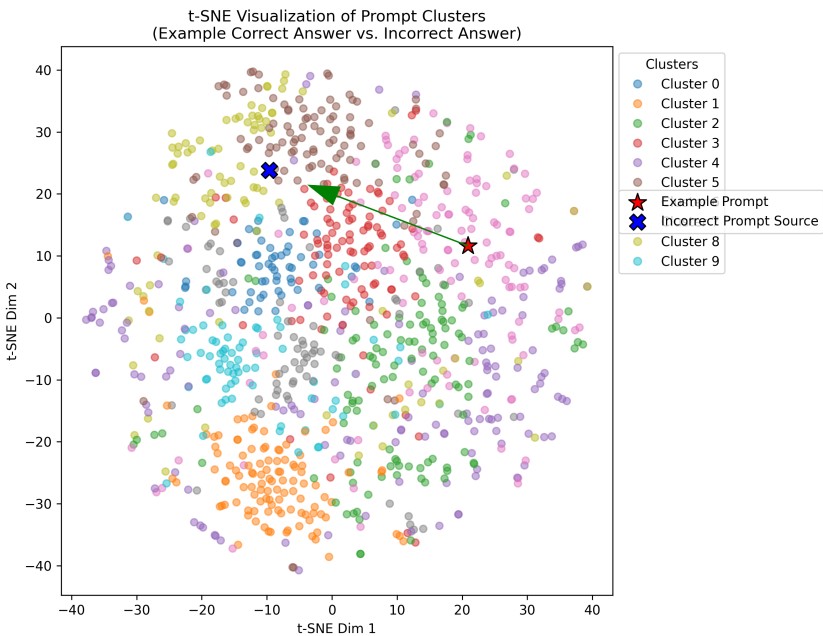

Figure 5: An example of selecting correct and incorrect prompt for a prompt and summarized answers.

```
{prompt_text}
ANSWER:
{summarized_answer}
Question:  Is this answer relevant to the prompt, or is it
irrelevant??
Please respond with exactly one word:  "relevant" or "irrelevant".
```

