# OpenReview forum: "ENHANCING DIVERSITY AND NOVELTY IN TEXT GENERATION VIA MULTI-VIEW EMBEDDINGS"
_ICLR.cc/2025/Workshop/AgenticAI — ICLR 2025 Workshop AgenticAI Poster_

### Official Review · Reviewer_zBNX · 2025-03-04
**Reviews of Submission 31**

**Rating:** 5
**Confidence:** 4

**Review:**

**Strengths**
- The paper introduces a model-agnostic method using multi-view embeddings (both text and image-based) that can be applied to various LLMs without architectural modifications, making it widely applicable across models.
- The study presents extensive experimental results across 469K generated outputs from various LLMs, demonstrating significant improvements.

**Weaknesses**
- While the paper explains the process of using image views, it doesn't clearly describe how relevant images are initially selected for a given prompt or how the quality of these images impacts performance.
- The paper would benefit from more direct comparisons with other techniques that aim to improve diversity in text generation (such as decoding strategies like nucleus sampling).
- Some practical aspects of implementing the multi-view approach remain unclear, such as computational overhead, latency impacts, and how it would integrate with existing applications.

---

### Official Review · Reviewer_p1QY · 2025-03-05
**Good Paper**

**Rating:** 7
**Confidence:** 4

**Review:**

The paper proposes a multi-view embedding approach to improve diversity and novelty in Large Language Model (LLM) text generation. Traditional LLMs often produce repetitive outputs due to their reliance on training data patterns. The multi-view embedding method addresses this by incorporating different perspectives from both textual and visual sources before generating responses. This approach is model-agnostic, meaning it can be used with any LLM without modifying its architecture. Additionally, the paper introduces a comprehensive evaluation framework that simultaneously measures diversity, novelty, and correctness in generated content. The study evaluates 469,000 generated responses, showing that multi-view embeddings significantly enhance response variety while maintaining relevance and accuracy, making this technique useful for tasks requiring creative and exploratory thinking.

Strengths of the Paper:
1. Improves diversity in LLM-generated text, reducing repetitive or overly deterministic responses.
2. Model-agnostic approach, making it compatible with different LLMs without changes.
3. Demonstrates significant improvements over standard prompting methods across multiple tasks.

Weaknesses of the Paper:
1. Limited to external context sources, which may not always be relevant or accurate.
2. No human evaluation of creativity, relying solely on automated metrics.

---

### Decision · Program_Chairs · 2025-03-05

Accept (Poster)